# Cyclone Freddy and its impact on maternal health service utilisation: Cross-sectional analysis of data from a national maternal surveillance platform in Malawi

Hussein H. Twabi[1,2,3]*, James Jafali[2,3], Leonard Mndala[2,3], Jennifer Riches[2,3], Edward J. M. Monk[3], Deborah Phiri[3], Regina Makuluni[3], Luis Gadama[1], Fannie Kachale[4], Rosemary Bilesi[5], Malangizo Mbewe[4], Andrew Likaka[4,6], Chikondi Chapuma[2,3], Moses Kumwenda[1,3], Bertha Maseko[3], Chifundo Ndamala[3], Annie Kuyere[3], Laura Munthali[3], Marc Y. R. Henrion[3,5], Chisomo Msefula[1], David Lissauer[2,3], Maria Lisa Odland[2,3,7,8]

1 Kamuzu University of Health Sciences, Blantyre, Malawi, 2 Institute of Life Course and Medical Sciences, University of Liverpool, Liverpool, United Kingdom, 3 Malawi-Liverpool-Wellcome Trust Programme, Blantyre, Malawi, 4 Ministry of Health, Lilongwe, Malawi, 5 Universidade Federal de Pernambuco, Recife, Brazil, 6 Liverpool School of Tropical Medicine, Liverpool, United Kingdom, 7 Department of Public Health and Nursing, Norwegian University of Science and Technology, Trondheim, Norway, 8 Department of Obstetrics and Gynaecology, St. Olavs University Hospital, Trondheim Norway

* husseintwabi@hotmail.com

**Data Availability Statement:** Permission and access was sought from the Malawi Ministry of Health and the Malawi-Liverpool-Wellcome Trust

## Abstract

*Climate change poses a significant threat to women's health in sub-Saharan Africa, yet the impact of climate change on maternal health is rarely reported in the region. Using an existing Maternal Surveillance Platform (MATSurvey), we estimated the immediate impact of Cyclone Freddy on maternal health care service indicators in Malawi. We analysed facility-level data for pregnant women up to 42 weeks post-partum using the national MATSurvey database. We compared incidences of service utilisation before (1 January to 19 February 2023) and after (20 February to 30 March 2023) the cyclone using a negative binomial regression approach. Between 1 January and 30 March 2023, a total of 37,445 live births, 50,048 antenatal clinic attendances, 23,250 postnatal clinic attendances, 84 maternal deaths, and 1,166 neonatal deaths were recorded by 33 facilities in the MatSurvey database. There was an immediate reduction in service utilisation in the post-cyclone period, including the postnatal attendance per week (pre-cyclone median: 355.0 [IQR 279.0–552.0], post-cyclone median: 261.0 [IQR 154.3–305.5], RR 0.56 [95% CI 0.44–0.71, p <0.001]) and the antenatal attendance per week (pre-cyclone median: 860.0 [IQR 756.5–1060.0], post-cyclone median: 656.5 [IQR 486.5–803.3], RR 0.66 [95% CI 0.55–0.78, p <0.001]). Stratified analyses by geographical zones revealed a stronger reduction in postnatal clinic attendance in the Southwest (RR 0.50 [95% CI 0.29–0.85, p = 0.010]) and the North (RR 0.29 [95% CI 0.15–0.56, p <0.001]). Cyclone Freddy resulted in an immediate decline in utilisation of maternal health services in cyclone-affected regions in Malawi. We observe evidence of catastrophic climate events impacting on the healthcare of women and their babies. Policymakers, researchers, and health systems need to ensure that essential*

Programme for the data that was used in the analysis. Presently, we have not been given permission to share the data publicly. Requests for anonymised data, aggregated by facility, can be made by contacting the Malawi Ministry of Health and the Malawi–Liverpool–Wellcome Clinical Research Programme at dlissauer@mlw.mw.

**Funding:** This research was funded by the Bill and Melinda Gates Foundation (INV-004839) and, in part, by the Wellcome Trust (206545/Z/17/Z). DL is supported by a National Institute for Health and Care Research (NIHR) Global Health Professorship (NIHR300808), using UK aid from the UK Government to support global health research. The views expressed in this publication are those of the authors and not necessarily those of the NIHR or the UK Government. The funders had no role in study design, data collection, data analysis, data interpretation, writing of the report, or the decision to submit this paper for publication.

**Competing interests:** The Wellcome Trust has provided a Strategic Award to the Malawi–Liverpool–Wellcome Clinical Research Programme (206545/Z/17/Z) that, in part, covers the salary and operational costs of the Statistical Support Unit at the programme, led by MYRH. All other authors have no competing interests to declare.

women's health services are maintained during these events and improve measures to support service resilience in the face of climate change.

## Background

Maternal health in Sub-Saharan Africa remains a major challenge, despite a decline in maternal mortality over the past few decades [1]. The region has the highest maternal mortality ratio in the world, with an estimated 542 deaths per 100,000 live births in 2017 [2]. This means that a woman in sub-Saharan Africa is 20 times more likely to die from pregnancy-related causes than a woman in a developed country, even with improvements in sociodemographic status, healthcare access, family planning, and women's education [1].

Malawi, a landlocked country in South-Eastern Africa, has significant socio-economic challenges, including poverty, food insecurity, and limited access to healthcare and education [3, 4]. Maternal health in Malawi remains a critical concern, with a high maternal mortality ratio and limited access to quality healthcare services [4]. Cyclone Freddy was the longest-lasting category 5 cyclone recorded worldwide, and its effects were first experienced in Malawi from 19 February 2023, and again from 12 March 2023, after traversing the Indian Ocean for five weeks, devastating South-Eastern Africa in the coastal areas of Madagascar and Mozambique [5]. Cyclone Freddy severely impacted Malawi, through flooding, landslides, and infrastructural damage, in regions that are not yearly affected by disasters, especially of this magnitude (Fig 1B). The cyclone killed over a thousand people, and several thousand people were displaced, and essential services such as healthcare, water, and sanitation were disrupted [5].

Climatic disasters profoundly impact maternal health in Malawi and other resource-poor settings. Women are often disproportionately affected due to existing social disparities and heightened vulnerabilities during crises [8, 9]. The 2023 Tropical Cyclone Freddy in Malawi highlighted the mental health challenges faced by women in disaster-affected areas [8]. Similarly, the COVID-19 pandemic resulted in increased maternal deaths and decreased postnatal care visits in Malawi [10]. Disasters can reduce foetal growth in some women, although the effects on gestational age remain less clear [11]. The severity of disaster exposure significantly impacts maternal mental health, which may influence child development more profoundly than direct prenatal stress [11]. To address these challenges, comprehensive disaster preparedness plans should include psychological assessments and recovery strategies [8], with relief efforts focusing on the most exposed women [11].

Cyclones have a complex and interconnected relationship with climate change. Climate change, driven primarily by the increased concentration of greenhouse gases in the Earth's atmosphere, has been leading to a rise in sea surface temperatures [12, 13]. Warmer oceans provide the energy necessary for the formation and intensification of cyclones [12, 13]. As these storms move over warm waters, they absorb heat and moisture, fuelling their strength and potential for extreme weather events [12, 13]. Furthermore, climate change has also been associated with altered weather patterns and increased atmospheric moisture content, which can influence the intensity of cyclones [12–14]. Rising sea levels, another consequence of climate change, can exacerbate the damage caused by cyclones, as higher sea levels lead to more extensive storm surges and coastal flooding [12, 13]. All these factors highlight the undeniable connection between cyclones and climate change, making it imperative to address and mitigate the human-induced factors driving this climate shift to reduce the risks associated with cyclonic activity.

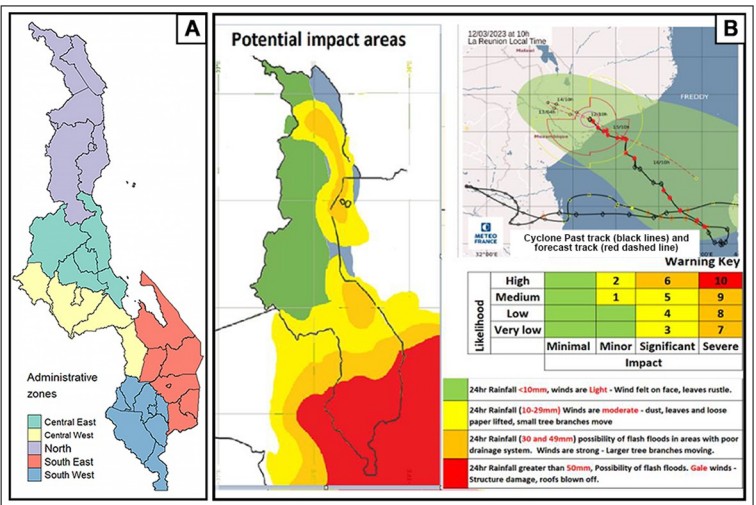

**Fig 1. Map of Malawi showing the administrative zones and the cyclone affected areas.** (A) Map of Malawi divided into the health administration zones (*map generated in R*, *shapefile source*: *humdata.org* [6]) (B) Heatmap of the potential impact regions of Cyclone Freddy in Malawi as of 12 March 2023 (*source*: *DoDMA 2023* [7]); The colours and the numbers in the risk matrix in Panel B represent the severity of the risk, combining the likelihood of a disaster occurring with the potential impact of that event; The cyclone past track denotes the historical path of the cyclone and the fortrack projects the path of the cyclone as of 12 March 2023.

Climate change is one of the most pressing challenges of our time [15]. The consequences are far-reaching and may significantly impact health, especially for vulnerable populations such as women and children in low-income settings [15, 16]. Rising temperatures and extreme weather events, such as heatwaves and prolonged droughts, can exacerbate existing health disparities and increase the prevalence of infectious diseases, posing additional risks to maternal health in the region. However, disasters and climate-change related events and their effects on maternal healthcare utilisation and health outcomes are poorly documented in low-resource settings.

Women in low-income countries, rural areas, and marginalised communities already face barriers in accessing essential reproductive and maternal healthcare services due to a lack of infrastructure, resources, and trained healthcare professionals [17]. Climate change-related disasters, like Cyclone Freddy, further strain already fragile healthcare systems, making it even more difficult for women to access healthcare services. Evidence of the effects of climate change and natural disasters on health systems are crucial to inform policy changes that incorporate climate resilience in their frameworks. Additionally, understanding gender-specific health consequences of climatic events is essential for developing effective disaster relief strategies and interventions that improve emergency response efforts, and ultimately reduce the burden on women's health in disaster-prone regions [9, 18]. We thus investigated the immediate effects of cyclone Freddy on maternal and reproductive health care service indicators using an existing national maternal surveillance platform in Malawi.

## Methods

### Study design and participants

The national maternal surveillance platform (MATSurvey) has been described in detail previously [19]. The online surveillance platform was initially established as a collaboration between Malawi's Ministry of Health and the Malawi-Liverpool-Wellcome Clinical Research

Programme (MLW) to understand and mitigate the effects of COVID-19 in pregnant and recently pregnant women (up to 42 days post-delivery). Additionally, the platform serves to strengthen Malawi's Maternal Death Surveillance and Response (MPDSR) structures.

MatSurvey has been implemented across 33 sites in Malawi, including all district (27 hospitals) and central hospitals (four hospitals) in Malawi. The platform collects individual-level data on all maternal deaths, as well as aggregate data on maternal near-miss cases and facility-level data such as numbers and types of births, staffing and resource availability. In this study, the data available extended to 30 March 2022. As such, to avoid seasonal heterogeneity, we have included aggregate facility-level data for women in the seven weeks leading up to the cyclone (1 January 2023, to 19 February 2023) and the six weeks following the cyclone (19 February to 30 March 2023).

Data for this analysis were pre-aggregated by facility and entirely anonymized and made available to the authors by the authorization of the Malawi Ministry of Health and College of Medicine Research Ethics committee.

### Data collection

Routine clinical data were collected and uploaded to the MATSurvey platform daily using a mobile data collection tool (OpenDataKit version 1.21.0) by nurse midwives acting as "safe motherhood coordinators" for each site. Source documents for data collection include patient clinical notes, ward handover files, hospital, and theatre registers.

### Outcomes

Our aim was to assess the impact of Cyclone Freddy on utilisation of maternal and reproductive health services in affected areas of Malawi. As such, we examined pre- and post-cyclone numbers of antenatal visits, postnatal visits, Caesarean sections, staffing levels at facilities, numbers of family planning clinic visits and cervical cancer screenings. Due to the short duration of the post-cyclone period, we assumed that adverse clinical outcomes would not immediately change. Thus, we excluded clinical outcome indicators from our analysis.

### Statistical analysis

We analysed facility-level weekly aggregated data to compare service utilisation indicators (antenatal clinic attendance, postnatal clinic attendance, and family planning clinic attendance) before and after the cyclone. Firstly, we reported and compared summary statistics (median and interquartile range [IQR]) between the cyclone periods and applied time series plots to visualise weekly variations in service utilisation indicators between 2022 and 2023, highlighting the pre- and post-cyclone periods. A negative binomial regression approach was used to calculate rate ratios (RRs) and 95% confidence intervals (CIs) to demonstrate the association between each service indicator and period (pre- or post-cyclone). This model estimated rate ratios and 95% confidence intervals while accounting for potential confounding (by variations in geographical zones) and overdispersion. Geographical zones are defined by pre-existing administrative zones utilised by the Malawi Ministry of Health in the surveillance of maternal mortality in the country. These zones include the Southeast, Southwest, Central East, Central West, and the Northern zone–Fig 1A.

Briefly, the following negative binomial equation was applied to model the counts of each maternal service utilization indicator:

$$Y_i \sim NB(\mu_i, r)$$

$$log(\mu_i) = \beta_0 + \beta_1 \, Period + \beta_{2k} \, Zone$$

Where: $Y_i$ is the count of service utilization indicator for the $i$th observation, $\mu_i$ is the expected count of the service utilization indicator for the $i$th observation, $r$ is the dispersion parameter for the negative binomial distribution, $\beta_0$ is the intercept, the adjusted log of the rate of health care utilisation events in the pre-cyclone period, $\beta_1$ represents the average change in the logarithm of the event rate associated with the post-cyclone period such that a positive/negative value indicates an increase/decrease in utilization due to the cyclonic event (respectively), and $\beta_{2k}$ represents the average difference in the logarithm of the event rate associated with geographical zone K compared against the reference zone (Central West zone). A likelihood ratio test (LRT) was applied to test for potential interaction (effect modification) between the cyclone period and the geographical zone.

A Difference-in-Difference (DiD) test was conducted using an LRT to examine the difference in trends in service utilisation between 2022 and 2023. Two negative binomial models were constructed: a full model, which modelled the weekly trends in service utilization from 1 January to 30 March with an interaction term between the weekly trends and the year, as well as main effects, and a reduced model, which only included the main effects of the weekly trends in service utilisation and the year, but not their interaction. An LRT p-value of less than 0.05 was considered significant.

All the analyses were conducted in R statistical computing software (R version 4.3.0 (2023-04-21 –R Core Team (2023), https://www.R-project.org/)). P-values less than 0.05 were considered significant.

## Ethical considerations

The MATSurvey platform is a routine surveillance platform that collects aggregate facility-level count data and does not include individual level data. As such, the study neither involved any individual patients nor their respective data. The authors of the paper obtained permission to use the data for this analysis from relevant individuals from the Malawi Ministry of Health and the Malawi-Liverpool-Wellcome Trust Programme. This study was conducted in accordance with the ethical standards of the General Data Protection Regulation (GDPR). The original data collection was approved by the College of Medicine Research Ethics Committee.

## Role of the funding source

The funders had no role in study design, data collection, data analysis, data interpretation, writing of the report, or the decision to submit this paper for publication.

## Results

### Summary of service and outcome indicators during the study period

The total number of live births during the study period was 37,445, with 7,108 (19.0%) delivered via Caesarean section and 545 (1.5%) via instrumental delivery. The total numbers of maternal and neonatal deaths during the study window were 84 and 1,166, respectively. The maternity units of the facilities had a total number of 18 doctors, 29 clinical officers and 76

nurses on duty during the study window. The total antenatal and postnatal attendance during the study period was 50,048 and 23,250 women, respectively.

Table 1 presents the summary statistics and results from the negative binomial regression models for service and outcome indicators of maternal and neonatal health during the study period, aggregated from the eligible 33 health facilities. There was a reduction in the median number of antenatal and postnatal clinic attendees in the post-cyclone period. The median pre-cyclone antenatal clinic attendance was 860.0 (IQR 756.5–1060.0) compared to 656.5 (IQR 486.5–803.3) in the post-cyclone period (p<0.001). The median pre-cyclone postnatal clinic attendance was 355.0 (IQR 279.0–552.0) compared to 261.0 (IQR 154.3–305.5) in the post-cyclone period (p<0.001).

## Time trends

Fig 2 illustrates the trends in the various maternal health indicators examined from the Mat-Survey dataset. There was a marked decrease in the number of antenatal and postnatal clinic attendees in the post-cyclone period, most notably in the country's Southwestern region–Fig 2. There was also a reduction in the number of caesarean sections performed in facilities in the Southwest region. There was also a decrease observed in the family planning and cervical cancer screening indices in the Southwestern and Central Eastern zones. There was a significant difference in the trends in service utilisation indicators between 2022 and 2023 (difference-in-difference p-value of <0.001 for all indicators).

## Impact of Cyclone Freddy on maternal health attendance indicators

There was a significant association between the cyclone period and a drop in the service utilisation indicators–Table 1. The most significant effect was observed in postnatal clinic attendance (RR 0.56 [95% CI 0.44–0.71, p<0.001]). Upon stratification by zone, the zone with the most significant reduction in service utilisation in the post-cyclone period was the southwestern zone–Fig 3. This effect was most remarkable for the number of women accessing cervical cancer screening services (RR 0.41 [0.23–0.74, p = 0.001]), the number of women accessing family planning services (RR 0.44 [0.27–0.73, p = 0.003]), number of antenatal clinic attendees (RR 0.50 [0.34–0.75, p = 0.010]) and number of postnatal clinic attendees (RR 0.50 [0.29–0.85, p<0.001]).

Effects of the cyclone were also observed in the Central East, with the most significant reductions being observed in the number of available doctors (RR 0.36 [0.17–0.79, p = 0.010]),

**Table 1. Median service utilisation per week and crude association with the cyclone event.**

| Indicator | Pre-cyclone | Post-cyclone | | p-value |
|---|---|---|---|---|
| | Median (IQR) | Median (IQR) | RR (95% CI) | |
| Antenatal clinic attendance | 860.0 (756.5;1060.0) | 656.5 (486.5;803.3) | 0.66 (0.55;0.78) | <0.001 |
| Postnatal clinic attendance | 355.0 (279.0;552.0) | 261.0 (154.3;305.5) | 0.56 (0.44;0.71) | <0.001 |
| Family planning clinic | 805.0 (705.0;1010.5) | 647.0 (469.0;878.8) | 0.73 (0.61;0.87) | <0.001 |
| Cervical cancer screening | 320.0 (198.0;480.0) | 162.5 (103.5;328.0) | 0.63 (0.48;0.82) | <0.001 |
| Births | 712.0 (561.0;815.0) | 493.5 (401.0;670.3) | 0.72 (0.6;0.87) | <0.001 |
| Caesarean sections | 141.0 (99.0;160.5) | 99.5 (46.8;134.5) | 0.65 (0.54;0.78) | <0.001 |
| Instrument deliveries (vacuum or forceps) | 8.0 (5.0;11.0) | 4.5 (3.0;9.8) | 0.59 (0.4;0.85) | 0.005 |
| Available doctors in the maternity unit | 5.0 (3.0;8.0) | 3.0 (1.0;5.5) | 0.65 (0.51;0.82) | <0.001 |
| Available clinical officers in the maternity unit | 20.0 (17.0;22.5) | 14.5 (12.0;21.8) | 0.77 (0.66;0.9) | <0.001 |
| Available nurses working in the maternity unit | 47.0 (42.5;53.5) | 31.5 (17.3;42.0) | 0.63 (0.51;0.79) | <0.001 |

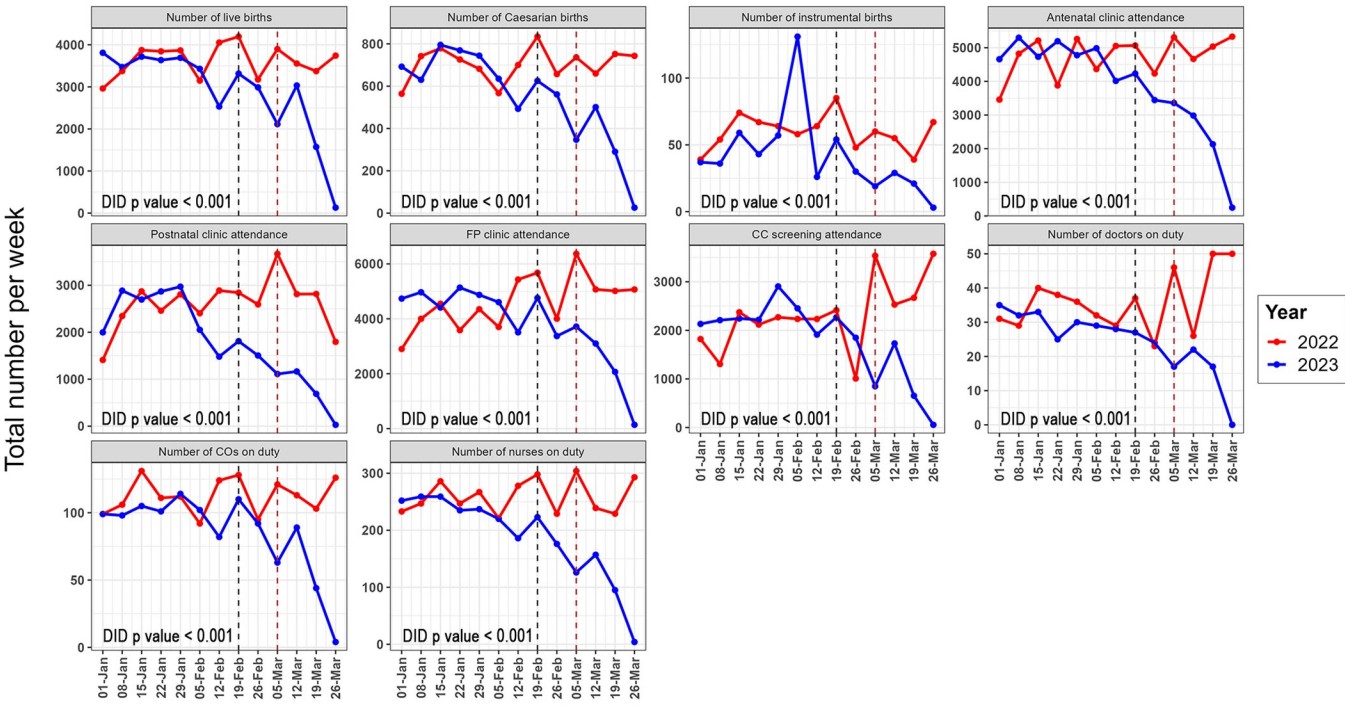

**Fig 2. Temporal trends in weekly maternal health service utilisation from 33 Malawian facilities from 1 January to 30 March for 2022 and 2023.** The grey dashed line corresponds to the day the cyclone reached peak intensity (19 February 2023) and the red dashed line corresponds to the day the cyclone made landfall in Malawi (12 March 2023). FP = family planning; CC = cervical cancer; CO = clinical officer; DID = difference-in-difference.

number of instrumental deliveries performed (RR 0.43 [0.20–0.94, p = 0.034]), and the number of postnatal clinic attendees (RR 0.53 [0.37–0.76, p<0.001]). A significant reduction in postnatal clinic attendance was also observed in the Northern Region of the country (RR 0.29 [0.15–0.56, p<0.001]). No effects were observed in the Central West and the Southeast zones in the five weeks post the cyclone.

## Discussion

In this study, we have shown a significant decrease in maternal and reproductive healthcare utilisation in the cyclone-affected regions of Malawi in the six weeks following Cyclone Freddy.

There needs to be more data on the impact of climate-related natural disasters on maternal and reproductive health service utilisation in sub-Saharan Africa. Previous studies in low- and middle-income countries have concurred mainly with our findings [20–22]. A review by Loewen et al. that included evidence from 13 studies from low and middle-income countries reported overall disruptions to service utilisation after natural disasters [23]. It is worth noting that 8 of the 13 studies included were from Africa but were on the effect of sexual and reproductive health services utilisation in the context of the Ebola outbreaks. A recent study in Bangladesh found that despite an overall lower usage of maternal healthcare services in flood-affected regions, no effect was observed in utilisation after a flood event [22]. It is possible that due to repeated exposures to such flooding events, the communities in these regions adapt to such conditions, resulting in different observed effects of each natural disaster event. Thus, these findings do not contradict our findings as the areas affected by the floods in our study

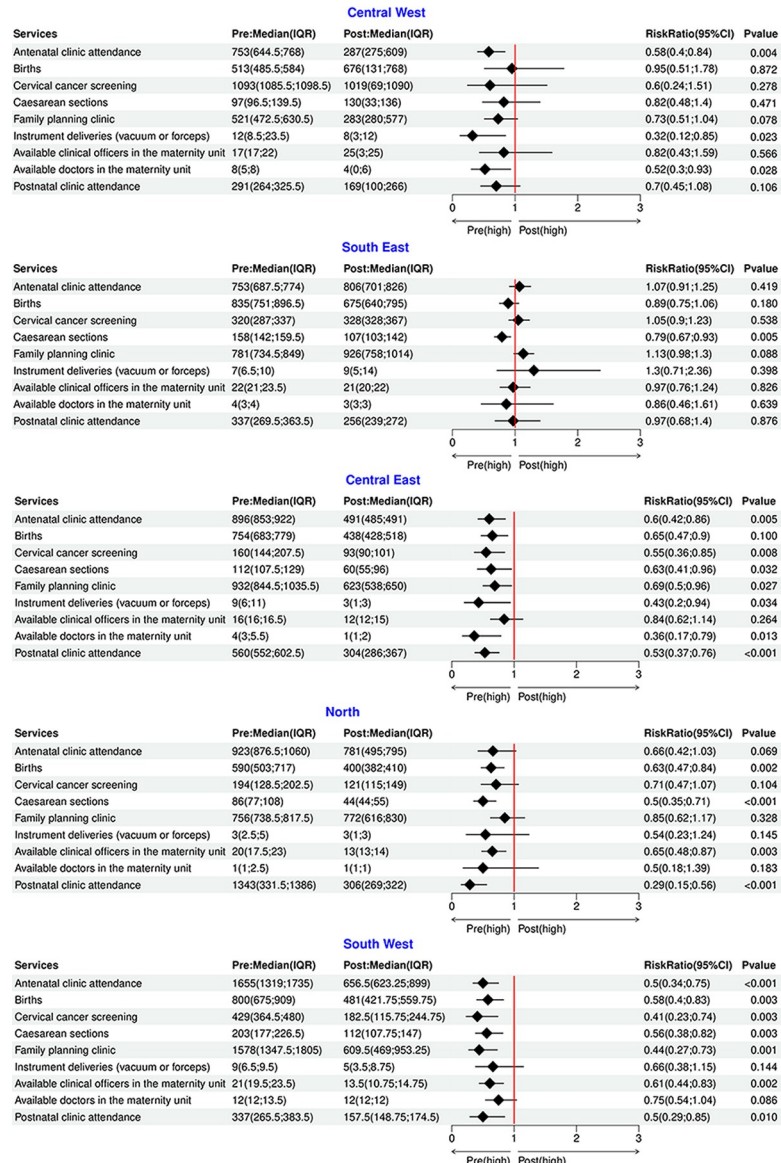

**Fig 3. Stratified negative binomial model of effect of cyclone period on maternal health service utilisation.**

were all but naive to disasters of such magnitude compared to the regions that experience the yearly monsoon floods of Bangladesh.

The most significant impact of Cyclone Freddy identified in our study was a reduction in postnatal clinic attendance, especially in the Southwestern, Central Eastern and Northern regions of Malawi, along with a decrease in the utilisation of other maternal and reproductive services. This could be because women are more likely to de-prioritise postnatal clinic attendance in times of crisis as they may view care after delivery as less critical. These findings underscore the importance of timely postnatal care, especially in crisis scenarios, given that this is when the majority of maternal deaths occur [24, 25].

Amidst these observations, it is crucial to consider the underlying factors that might contribute to the changes observed in maternal and reproductive healthcare utilisation following geoclimatic events. The complex interplay of socio-economic challenges, disrupted

infrastructure, and altered community dynamics during and after a natural disaster could decrease healthcare utilisation [22, 26]. Regional inequalities in maternal health care across Malawi exist, with wealth-based disparities in access to skilled birth attendance highest in the central region, followed by the north, and lowest in the south [27], potentially contributing to the effects observed in these regions that were further from the cyclone impact zone. Additionally, the Southern region has a higher number of facilities providing maternal health care services that the Central and Northern regions [28], thus the strength of the effects of the cyclone observed in the South confirms the magnitude of the cyclonic event experienced in 2023. Other factors that affect health access may have also contributed to the observed effects between the regions, including geographical terrain, road infrastructure and community and cultural dynamics. Communities facing the aftermath of such climatic events often experience difficulties accessing healthcare facilities due to damaged roads, overwhelmed health services, and limited resources. Moreover, factors like fear, anxiety, and a shift in priorities towards immediate survival might discourage individuals from seeking regular healthcare, particularly during the vulnerable postnatal period [8, 11, 29, 30]. Addressing these multifaceted influences demands a holistic approach beyond medical services alone, incorporating community engagement, targeted support systems, and effective disaster preparedness strategies.

Our study has some limitations. Firstly, we sought to ensure the timely reporting of these results given the urgency of the crisis and understanding its response. We therefore proceeded to the analysis with limited post-cyclone data, resulting in potential imprecision in the estimated effect. Though the use of a reduced pre-cyclone period helps to mitigate this effect and maximise the precision by which we could report estimated effect sizes. Furthermore, due to the limited post-cyclone period, we could not explore the impact of the cyclone on longer-term health outcome indicators, as effects on these indicators only become apparent after a significant time lag, thus resulting in an incomplete understanding of the impact of the cyclone on maternal health. Furthermore, the data lacks individual-level information that would allowing for stratification and adjusted analyses to examine the associations further. Nevertheless, the availability of data from all the regions in the country allowed for an informative and timely investigation of the immediate effects of a cyclonic disaster.

In conclusion, Cyclone Freddy resulted in an immediate decline in the reported utilisation of maternal and reproductive health services in both the cyclone-affected and unaffected regions of Malawi. This study indicates the immediate effects of such climatic events in low-income settings and underscores the need for disaster-resilient healthcare structures to better withstand the effects of climate change and climate-related disasters. Policies and interventions should thus be tailored to emphasise the critical role of maternal and reproductive health services within disaster response frameworks to ensure continuity and accessibility during and after such events. Further studies with extended observation periods are warranted to comprehensively examine the long-term effects of intense cyclones on maternal and child health outcomes in the region. Additionally, research should focus on the development and implementation of climate change adaptation strategies in healthcare systems to enhance their resilience and capacity to provide uninterrupted care during climatic disasters.

## Patient and public engagement

The Kamuzu University of Health Sciences, in conjunction with the Malawi-Liverpool-Wellcome Trust Clinical Research Programme, is continually involved in community engagement. The institutions formed a community advisory board (CAB), who act as liaisons between the researchers and the community at large. The development of the original maternal health surveillance platform was directly informed by discussion with the community representatives,

wherein several health priorities for the community were identified. The current study was informed by unstructured interactions, during volunteer medical outreach activities, with pregnant and breastfeeding women whose lives were disrupted by the Cyclone Freddy. The CAB also functions as a platform through which research findings can be fed back to the community.

## Supporting information

**S1 Fig. Weekly trends in maternal health utilisation indicators in 2023 stratified by zones.**
(TIF)

**S2 Fig. Autocorrelation Function (ACF) Plot demonstrating autocorrelation in the residuals.**
(TIFF)

**S3 Fig. Plot of residuals against fitted values for the negative binomial models.**
(TIF)

**S1 Table. Overdispersion testing for fitted poisson and negative binomial models for trends in uptake of maternal services pre- and post-Cyclone Freddy.**
(DOCX)

## Author Contributions

**Conceptualization:** Hussein H. Twabi, Maria Lisa Odland.

**Data curation:** Leonard Mndala.

**Formal analysis:** Hussein H. Twabi, James Jafali.

**Methodology:** James Jafali.

**Supervision:** Marc Y. R. Henrion, Chisomo Msefula, David Lissauer, Maria Lisa Odland.

**Writing – original draft:** Hussein H. Twabi, Maria Lisa Odland.

**Writing – review & editing:** Hussein H. Twabi, James Jafali, Leonard Mndala, Jennifer Riches, Edward J. M. Monk, Deborah Phiri, Regina Makuluni, Luis Gadama, Fannie Kachale, Rosemary Bilesi, Malangizo Mbewe, Andrew Likaka, Chikondi Chapuma, Moses Kumwenda, Bertha Maseko, Chifundo Ndamala, Annie Kuyere, Laura Munthali, Marc Y. R. Henrion, Chisomo Msefula, David Lissauer, Maria Lisa Odland.

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
