## [Decision Letter · Decision Letter 0]

10 Jun 2024

PGPH-D-24-00832

Cyclone Freddy and its Impact on Maternal Health Service Utilisation: Cross-sectional Analysis of Data from a National Maternal Surveillance Platform in Malawi

Dear Dr. Twabi,

Thank you for submitting your manuscript to PLOS Global Public Health. After careful consideration, we feel that it has merit but does not fully meet PLOS Global Public Health’s publication criteria as it currently stands. Therefore, we invite you to submit a revised version of the manuscript that addresses the points raised during the review process.

The reviewers agree that the paper addresses an important and pertinent issue. However, the presentation of the work could be improved and the study context could be better conveyed. Reviewer 1 has some suggestions for the Background and Discussion. In addition, it would be good to provide some model validation.

We look forward to receiving your revised manuscript.

Kind regards,

Michele Nguyen

Academic Editor

Journal Requirements:

1. Please send a completed 'Competing Interests' statement, including any COIs declared by your co-authors. If you have no competing interests to declare, please state "The authors have declared that no competing interests exist". Otherwise please declare all competing interests beginning with the statement "I have read the journal's policy and the authors of this manuscript have the following competing interests:"

2. Tables should not be uploaded as individual files. Please remove these files and include the Tables in your manuscript file as editable, cell-based objects. For more information about how to format tables, see our guidelines:

https://journals.plos.org/globalpublichealth/s/tables

Additional Editor Comments (if provided):

Please check for typological errors: e.g. RR 29 for the risk ratio of postnatal clinic attendance in the North should be 0.19. (p.3, Findings), "calculated" should be "calculate" and Figure 2A should be Figure 1A (p.6, 3rd paragraph).Please check for notation consistency e.g. in how you quote dates on p.4.p.5, 4th paragraph: There are seven, not six weeks, from 1st January 2023 to 19th February 2023. You mentioned six weeks (19th February to 30th March 2023) after the cyclone but on p.9 and the first paragraph of the Discussion, you mentioned fix weeks following the cyclone.p.6, 2nd paragraph: Would it be more accurate to say "utilisation of maternal and reproductive health services" rather than "access"?p.6, negative binomial equation: Please provide some model validation (e.g. checks for temporal correlation in residuals) in the Supplementary Material.p.7: Please write the bulletpoints as text.p.8, 3rd paragraph: Could you clarify how the difference-in-difference p-values were calculated? What 2022 and 2023 dates were used to compute the differences in service indicator values?p.10, 2nd paragraph: It would be useful to add the context ("the areas affected by the floods in our study were all but naive to disasters of such magnitude compared to the regions that experience the yearly monsoon floods of Bangladesh") to the Background on p.4.p.10, 4th paragraph: It is mentioned that "the complex interplay of socio-economic challenges, disrupted infrastructure, and altered community dynamics during and after a natural disaster could decrease healthcare utilisation". Could you comment on this more in the light of the varying impact observed in the different zones (Central East, Central West, North, South East and South West)?p.11, 2nd paragraph: It is mentioned that the lack of individual-level information prevents the further examination of associations. It may be possible to use zone-level data on e.g. the socio-economic indicators and transport system.Fig. 1B: Where was this figure referenced in the main paper? What do the colours and numbers in the warning key mean? Is the cyclone past track the actual observed track? When was the forecast track produced?Fig. 3: The first subfigure provides the same information as Table 1. You could either split the figure into two and increase the size of the figure so that the numbers and text are easier to read, or present the rest of the subfigures as tables (similar to Table 1).

Reviewers' comments:

Reviewer's Responses to Questions

**Comments to the Author**

1. Does this manuscript meet PLOS Global Public Health’s publication criteria? Is the manuscript technically sound, and do the data support the conclusions? The manuscript must describe methodologically and ethically rigorous research with conclusions that are appropriately drawn based on the data presented.

Reviewer #1: Partly

Reviewer #2: Yes

2. Has the statistical analysis been performed appropriately and rigorously?

Reviewer #1: I don't know

Reviewer #2: Yes

3. Have the authors made all data underlying the findings in their manuscript fully available (please refer to the Data Availability Statement at the start of the manuscript PDF file)?

Reviewer #1: Yes

Reviewer #2: Yes

4. Is the manuscript presented in an intelligible fashion and written in standard English?

Reviewer #1: Yes

Reviewer #2: Yes

5. Review Comments to the Author

Reviewer #1: It is a valuable study but following comments need to be considered:

-the background needs to be completed using more valid references and related evidence. The authors need to highlight the problem statement and why they designed this project to address it. what is the added value and the necessity of this research since we know that disasters effects women' s health and I have investigated and reported since years ago. What are the innovations of your research?

-mention the ethical code

-discussion needs to be improved by more interpretation of your findings comparing to other similar studies conducted in different settings.

- in the conclusion section, suggest more practical suggestions for your audience and interested stakeholders as well as more suggestions for further research

Reviewer #2: The manuscript is well written with a good and appealing title. This is an important topic given the current climate changes, emerging and re-emerging diseases, pandemics, disasters and other public health emergencies. It reflects our level of preparedness and readiness to detect, prevent and respond to disasters and other public health emergencies. It provides insight on how to handle essential health services on reproductive, maternal and child health during disaster or public health emergency response.

It is technically sound with data supporting the conclusion. The methodology and ethical considerations are appropriate and rigorous. However, it would have been better to establish whether there were any changes of measured indicators trends for similar comparable time frame in the past 3 or 5 years.

6. PLOS authors have the option to publish the peer review history of their article (what does this mean?). If published, this will include your full peer review and any attached files.

**Do you want your identity to be public for this peer review?** For information about this choice, including consent withdrawal, please see our Privacy Policy.

Reviewer #1: No

Reviewer #2: No

---

## [Editor Report · Decision Letter 1]

2 Aug 2024

Cyclone Freddy and its Impact on Maternal Health Service Utilisation: Cross-sectional Analysis of Data from a National Maternal Surveillance Platform in Malawi

PGPH-D-24-00832R1

Dear Dr. Twabi,

We are pleased to inform you that your manuscript 'Cyclone Freddy and its Impact on Maternal Health Service Utilisation: Cross-sectional Analysis of Data from a National Maternal Surveillance Platform in Malawi' has been provisionally accepted for publication in PLOS Global Public Health.

Best regards,

Michele Nguyen

Academic Editor